# Geometry-Embedded Neural Networks Cone Beam CT Reconstruction for Arbitrary Scanning Trajectories

**Nele Blum**[1] [iD]                                    NELE.BLUM@IMTE.FRAUNHOFER.DE

**Max Stickel**[1]                                       MAX.STICKEL@IMTE.FRAUNHOFER.DE

**Moritz Schaar**[1]                                     MORITZ.SCHAAR@IMTE.FRAUNHOFER.DE

**Thorsten M. Buzug**[1,2]                               THORSTEN.BUZUG@IMTE.FRAUNHOFER.DE

**Maik Stille**[1]                                       MAIK.STILLE@IMTE.FRAUNHOFER.DE

[1] *Fraunhofer Research Institution for Individualized Medical Technology and Engineering IMTE, Lübeck, Germany* [2] *Institute for Medical Engineering, University of Lübeck, Lübeck, Germany*

**Editors:** Under Review for MIDL 2026

## Abstract

Flexible scanning trajectories for CBCT systems offer enhanced imaging capabilities but pose challenges for conventional reconstruction algorithms. We present a novel deep learning-based reconstruction framework that explicitly integrates trajectory geometry information through differentiable back projection operators embedded within the network architecture. Our approach comprises three cascaded networks: a preprocessing network for projection filtering, an encoder-decoder reconstruction network with multi-scale back projection operators, enabling transition from projection to image domain, and a postprocessing network for final refinement. Evaluation on 2,000 test objects demonstrates substantial improvements over conventional filtered back projection (FBP), achieving an MSE of $4.8e^{-3}$ (vs. $2.4e^{-2}$ for FBP) and SSIM of 0.96 (vs. 0.54 for FBP).

**Keywords:** CT Reconstruction, Deep Learning, Non-Circular Scanning Trajectories.

## 1. Introduction

Cone-beam computed tomography (CBCT) systems enable high-resolution 3D imaging with low radiation exposure. Traditional circular or helical scanning trajectories produce excellent results in standard applications but encounter fundamental limitations in constrained environments, such as surgical or interventional procedures. Advances in robotics enable more flexible imaging systems capable of executing non-circular scanning trajectories, thereby addressing challenges including improved target region coverage with optimal dose application, artifact reduction, and enhanced image quality (Hatamikia et al., 2022). However, conventional analytical reconstruction algorithms often cannot accommodate modified trajectories without complex adaptations, while iterative methods require significantly longer reconstruction times that are unsuitable for time-critical applications. Deep learning methods offer a promising alternative by integrating convolutional neural networks into the reconstruction pipeline (Russ et al., 2022; Ye et al., 2025; Zhou et al., 2025), achieving comparable results to iterative methods in significantly shorter times. However, trajectory geometry information is rarely directly integrated into the learning process, and approaches incorporating trajectory information through deconvolution (Russ et al., 2022) remain susceptible to deviations between actual and assumed trajectories. We present a novel methodology for reconstructing CBCT data acquired along flexible, non-circular trajectories. Our method explicitly incorporates the scanning trajectory through integrated backprojection operators embedded within the network architecture, enabling highly flexible application across diverse acquisition geometries.

## 2. Methods

Figure 1 shows the training framework consisting of three components. First, the projection data are filtered by a preprocessing network. In parallel, the original acquisition data are processed by a second network, consisting of an encoder-decoder architecture featuring skip connections analogous to a U-Net, with embedded backprojection operators. Hence, the encoder operates in the projection domain while the decoder processes the converted data in the image domain. The backprojected preprocessed data, together with the output from the second network and a simple backprojection, are subsequently fed into a third postprocessing network. The backprojection layers were defined using the Parallelproj package (Schramm and Thielemans, 2024), which provides differentiable forward and back projectors for tomographic reconstruction. The geometry of the operators was flexibly

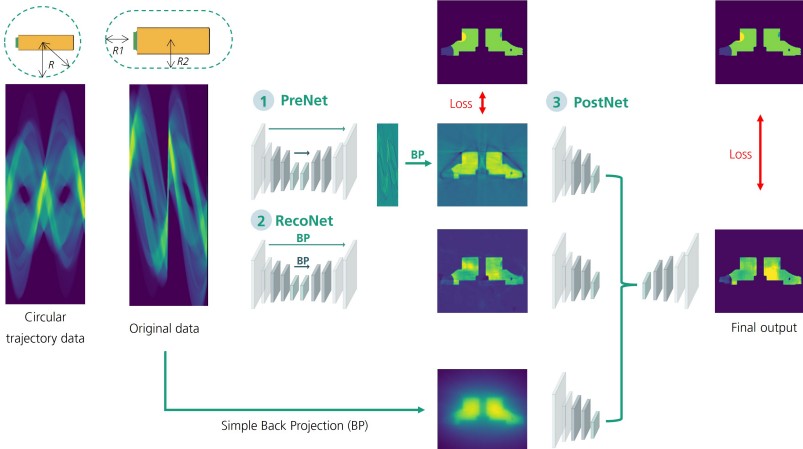

Figure 1: The preprocessing network (PreNet) filters the projection data, while the reconstruction network (RecoNet) employs an encoder-decoder architecture with embedded backprojection operators, transfering the projection data directy in the image domain. The postprocessing network (PostNet) combines both outputs, and a simple backprojection to generate the final reconstruction.

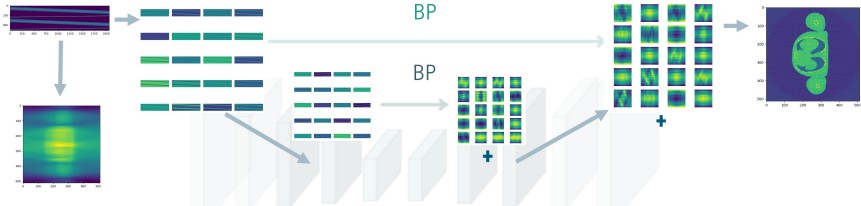

Figure 2: RecoNet with multi-scale backprojection operators. Five skip connections incorporate backprojection operators with varying geometrical resolutions: the original full-resolution and four progressively downsampled (factors of 2, 4, 8, and 16).

implemented so that it can be directly derived from any kind of detector and source position data and, therefore, any trajectory. In addition to the original geometry, downsampled geometries were employed in the intermediate layers of the RecoNet (Figure 2), whereby both the number of angular steps and the detector dimensions were reduced by factors of

2, 4, 8, and 16, resulting in a total of five skip connections, including one in the bottleneck of the network. To ensure a high degree of variation among different objects for training the networks, the Objaverse V1.0 (Deitke et al., 2023) library was utilized and modified by adding additional 3-D volumes with different attenuation values within the original object volume. Over 8,000 objects were initially used for training and validation of the networks, with an additional 2,000 objects for testing. Geometry variations were simulated using the ITK-RTK framework (Rit et al., 2013) by integrating a 20 mm translation at zero and 180 degrees (see Figure 1, SID: 150 mm (R1)/160 mm (R2), SDD: 600 mm). Training was conducted with a batch size of 4 and a learning rate of 0.0001 using an adam optimizer and MSE-Loss.

## 3. Results and Discussion

Figure 3 presents example objects from the test data set, displaying reconstructions of the PreNet, followed by the RecoNet and PostNet compared with the reference phantom. While the PreNet still exhibits pronounced artifacts at object boundaries and the images appear slightly blurred, the RecoNet demonstrates sharper structural features with minor residual artifacts. These are subsequently eliminated by the PostNet. Figure 4 shows quantitative results for all methods. Again, the final reconstruction from the PostNet shows substantial improvements.

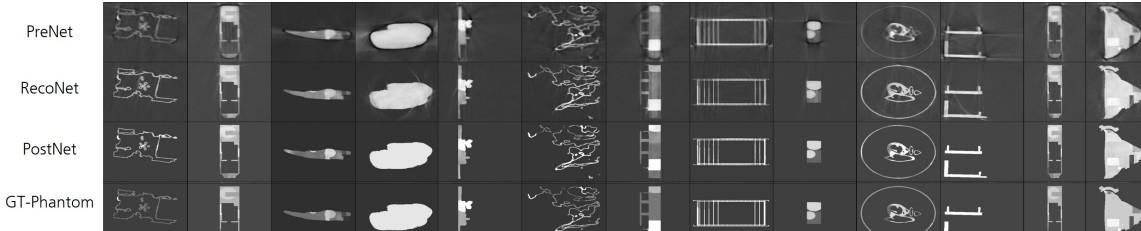

Figure 3: Example reconstructions from the test dataset.

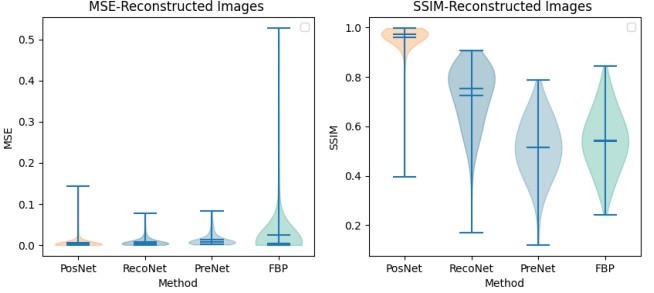

| Method | MSE | SSIM |
|--------|-----|------|
| FBP | 0.02423 | 0.5423 |
| PreNet | 0.01249 | 0.5165 |
| RecoNet | 0.00798 | 0.7262 |
| PostNet | **0.00480** | **0.9606** |

Figure 4: Violin plot and quantitative evaluation of reconstruction methods.

## 4. Conclusion

Our method achieves substantial improvements over conventional approaches while maintaining flexibility. Future work will validate the approach with clinical data, investigate alternative reconstructions, and assess robustness for different real-world trajectories.

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
