# OpenReview forum: "Geometry-Embedded Neural Networks Cone Beam CT Reconstruction for Arbitrary Scanning Trajectories"
_MIDL.io/2026/Short_Papers — MIDL 2026 - Short Papers Poster_

### Official Review · Reviewer_EGyn · 2026-04-23
**strong short paper limited to synthetic results**

**Rating:** 4
**Confidence:** 3

**Review:**

The paper’s main strengths are that it tackles a highly relevant CBCT problem, uses a physics-informed architecture that embeds acquisition geometry directly through differentiable backprojection operators, and shows a clear, intuitive design with strong synthetic gains over FBP. The multi-scale geometry-aware reconstruction idea is particularly appealing because it aligns well with how projection data should be mapped into image space. Its main weaknesses are that the evidence is still limited to synthetic experiments, the baselines are relatively weak, and the claim of handling “arbitrary trajectories” is broader than what is actually validated in the paper.

**Summary:**

This paper proposes a geometry-embedded deep cone beam CT reconstruction method for non-circular trajectories, where differentiable backprojection operators derived from the actual source-detector geometry are built directly into the network. The architecture combines a projection-domain preprocessing network, a U-Net-like reconstruction network with multi-scale backprojection skip connections, and a postprocessing network for refinement. On synthetic data with simulated trajectory variation, the method substantially outperforms standard FBP in both MSE and SSIM.

**Strengths:**

- CBCT reconstruction for flexible or non-circular trajectories is an important application
- Geometry is embedded through differentiable backprojection operators rather than treated only implicitly.
- Strong synthetic results versus FBP, with both quantitative and qualitative improvements.
- Clearly written paper

**Weaknesses:**

- Validation is limited to synthetic data
- Baseline methods are not very strong
- The title/claim of “arbitrary trajectories” is broader than what is experimentally demonstrated.
- More ablation is needed to isolate the value of the embedded multi-scale backprojection design.

**Justification Of Rating:**

The idea is compelling but the experimental validation is still limited relative to the breadth of the arbitrary trajectory claim.

---

### Decision · Program_Chairs · 2026-05-08

Accept (Poster)